# Evasion of serum antibodies and complement by *Salmonella* Typhi and Paratyphi A

**Fermin E. Guerra**[1], **Joyce E. Karlinsey**[2], **Stephen J. Libby**[1], **Ferric C. Fang**[1,2*]

**1** Department of Laboratory Medicine and Pathology, University of Washington, Seattle, Washington, United States of America, **2** Department of Microbiology, University of Washington, Seattle, Washington, United States of America

\* fcfang@uw.edu

## Abstract

Nontyphoidal and enteric fever serovars of *Salmonella enterica* display distinctive interactions with serum antibodies and the complement system, which initiate the host immune response to invading microbes. This study examines the contributions of lipopolysaccharide O-antigen (O-ag) and the *S.* Typhi Vi polysaccharide capsule to serum resistance, complement activation and deposition, and immunoglobulin (Ig) binding in nontyphoidal *S. enterica* serovar Typhimurium and the enteric fever serovars *S.* Typhi and *S.* Paratyphi A. Although all three serovars are resistant to serum killing, *S.* Typhi and *S.* Paratyphi A exhibit lower levels of Ig binding, complement binding and complement activation compared to *S.* Typhimurium. In *S.* Typhimurium, WzzB-dependent long O-antigen (L O-ag) production with 16-to-35 repeating O-ag units, and FepE-dependent very long O-antigen (VL O-ag) production with over 100 repeating O-ag units, are required for serum resistance but do not prevent IgM binding or complement deposition. *S.* Typhi lacks VL O-ag, but its production of Vi capsule inhibits IgM binding and complement deposition, while acting in concert with L O-ag to resist serum killing. In *S.* Paratyphi A, L O-ag production is deficient due to a hypofunctional WzzB protein, but this is compensated by greater quantities of VL O-ag, which are required for serum resistance. Restoration of WzzB function by exchange with the *S.* Typhimurium or *S.* Typhi *wzzB* alleles can restore L O-ag production in *S.* Paratyphi A but decreases VL O-ag production, resulting in increased IgM binding. Replacement of the *S.* Paratyphi A O2-type polysaccharide with the *S.* Typhi O9 polysaccharide further increases IgM binding of *S.* Paratyphi A, which enhances complement activation but not complement deposition. Lastly, a gene duplication of *rfbV* in *S.* Paratyphi A is necessary for higher levels of VL O-ag and resistance to complement deposition and antibody binding. Collectively, these observations demonstrate fundamental differences between nontyphoidal and enteric fever *Salmonella* serovars in their interactions with innate immune effectors. Whereas nontyphoidal *S.* Typhimurium elicits, exploits and withstands the host acute inflammatory response, the enteric fever serovars *S.*

**Data availability statement:** All data are within the manuscript and supporting files.

**Funding:** FEG received support (including salary support) from training grants T32 AA007509 and AI055396, and FCF was supported by R01 AI160130 (including partial salary support for all authors), all from the National Institutes of Health. The funders had no role in study design, data collection and analysis, decision to publish, or preparation of the manuscript.

**Competing interests:** The authors have declared that no competing interests exist.

Typhi and *S.* Paratyphi A evade it by limiting antibody recognition and complement activation and deposition.

---

## Author summary

Enteric fever acquired by ingestion of food or water contaminated with *Salmonella* Typhi or Paratyphi A is a significant cause of morbidity and mortality in low- and middle-income countries. In this study, we dissect the different mechanisms by which these bacteria avoid binding by serum proteins that are required for the initiation of protective immune responses. This contrasts with the nontyphoidal serovar *Salmonella* Typhimurium, a common cause of gastroenteritis, which does not avoid complement activation nor antibody binding but instead exploits host inflammation. The *Salmonella* Typhi Vi capsule, a major virulence determinant, prevents complement deposition and decreases antibody recognition. *Salmonella* Paratyphi A, which also causes enteric fever, does not contain Vi capsule but instead relies on increased very long O-antigen and decreased long O-antigen production to avoid complement deposition and antibody recognition. Our observations provide new insights into the mechanisms responsible for the distinctive immunological features of human enteric fever and can inform the development of *Salmonella* vaccines that target the *Salmonella* cell envelope.

## Introduction

*Salmonella enterica* is an enteric Gram-negative pathogen that causes disease in humans with clinical manifestations ranging from asymptomatic carriage to self-limited acute gastroenteritis or life-threatening systemic infection [1]. Antisera have long been used to subdivide *Salmonella* into >2,500 serovars based on specific lipopolysaccharide (LPS) and flagellar signatures, although only a few of these are prominent etiologic agents of human disease with distinctive clinical features [2]. Ingestion of nontyphoidal *S. enterica* serovar Typhimurium (*S.* Typhimurium) in contaminated food typically causes symptoms of acute gastroenteritis following a brief incubation period of 12–36 hours, with occasional extraintestinal complications that are particularly seen in individuals with immunosuppressive conditions [3]. In contrast, the enteric fever *Salmonella* serovars Typhi or Paratyphi A (*S.* Typhi or *S.* Paratyphi A) exhibit a more prolonged incubation period, typically lasting from 7 to 14 days, followed by fever, bacteremia, and systemic illness [4]. Complications of enteric fever include gastrointestinal bleeding, intestinal perforation, and encephalopathy, which is associated with high mortality, and some infected individuals become chronic asymptomatic carriers despite antibiotic treatment [4–6]. Salmonellosis caused by nontyphoidal serovars account for over 1 million infections in the United States annually, while enteric fever serovars are estimated to cause between 12 and 27 million infections and 200,000 deaths globally each year [7,8]. Enteric fever is

becoming increasingly difficult to treat due to the prevalence of multidrug-resistant strains [9]. Vaccines against *S.* Typhi are available but only partially protective and require booster doses, and currently there are no vaccines available for *S.* Paratyphi A or *S.* Typhimurium [10,11].

Decades of research have pointed to fundamental differences in the interaction of nontyphoidal and enteric fever *Salmonella* serovars in their interaction with the host immune system. Nontyphoidal serovars such as *S.* Typhimurium elicit an intense acute inflammatory response in the intestine, which is exploited by the bacterium to outcompete the commensal microbiota and facilitate transmission to other hosts by the production of inflammatory diarrhea [12–15]. In contrast, the enteric fever serovar *S.* Typhi employs a variety of mechanisms to avoid the stimulation of an acute inflammatory response, allowing it to persist within mononuclear cells and disseminate to extraintestinal sites [16–21]. Thus, while all *Salmonella* serovars share a common requirement to withstand host immune mediators, nontyphoidal and enteric fever serovars differ in whether to exploit or evade innate immunity.

The ability of *Salmonella* to cause invasive disease and bloodstream infection requires resistance to serum bactericidal mechanisms primarily mediated by the complement system, a complex surveillance system that bridges innate and adaptive immunity through the classical pathway of complement activation [22]. The classical pathway is defined by immunoglobulin G (IgG) or M (IgM) binding to antigen, which induces a conformational change to recruit the complement protein C1q and induce formation of the C3 convertase C4b2b, also known as C4b2a [23]. The C3 convertase is covalently bound to the bacterial surface or to the antigen recognition complex and cleaves complement protein C3 into the anaphylatoxin C3a and the C3b fragment. An internal thioester bond in C3b serves as the site of nucleophilic attack by hydroxyl and amino groups, which covalently link C3b to the bacterial surface [24,25]. When C3b binds to the C3 convertase C4b2b, the C5 convertase C4b2bC3b preferentially cleaves C5 into C5a and C5b. Complement fragment C5b forms a lipophilic complex with C6 and C7 to initiate insertion into targeted bacterial membranes. Sequential recruitment of C8 and C9 polymerization completes formation of the membrane attack complex (MAC), an 11 nm wide β-barrel pore on the bacterial membrane that leads to lysis and death [26,27]. In addition to mediating bacterial lysis, the anaphylatoxins C3a and C5a recruit inflammatory cells to the site of infection, and bound C3b promotes opsonophagocytosis of bacteria by phagocytic cells bearing complement receptors.

Lipopolysaccharide chains attached to the outer leaflet of the outer membrane of *Salmonella enterica* are exposed to the extracellular environment and provide resistance to the lytic action of complement [28–33]. The lipopolysaccharide of *Salmonella enterica* is composed of a hydrophobic membrane-inserted lipid A domain that is attached to a hydrophilic nonrepeating inner and outer core oligosaccharide [28]. The outer core is attached to O-ag repeating units, which in the *Salmonella* serovars Typhimurium, Typhi, and Paratyphi A share a common backbone composed of (→2)-D-mannose-(α1→4)-L-rhamnose-(α1→3)-D-galactose-(α1→) [34]. However, the distinct *S.* Typhimurium O4, *S.* Paratyphi A O2, and *S.* Typhi O9-antigens result from the (α1→3) linkage of D-abequose, D-paratose, or D-tyvelose, respectively, to D-mannose in the backbone. The extent of O-ag repeating units attached to the lipid A-core is controlled by the chain length determinants WzzB and FepE. WzzB-dependent L O-ag contains 16-to-35 O-ag repeating units, while FepE-dependent VL O-ag contains >100 repeating units [30,35,36]. In *S.* Typhi, *fepE* is a pseudogene, and the Vi capsule encoded by *Salmonella* pathogenicity island-7 (SPI-7) is produced rather than VL O-ag [37,38].

The roles of L and VL O-ag and the Vi capsule in serum resistance have been studied in *S.* Typhimurium and *S.* Typhi, and these structures are important vaccine targets. Previous studies have shown that WzzB-dependent L O-ag confers resistance to serum killing in *S.* Typhimurium and *S.* Typhi [31–33,39,40]. The *S.* Typhi Vi capsule also confers resistance to serum killing, but FepE-dependent VL O-ag is not required for serum resistance in *S.* Typhimurium [32,39]. In contrast, the roles of L and VL O-ag in *S.* Paratyphi A serum resistance are less clear. Hiyoshi et al. assessed the influence of VL O-ag on the phagocyte oxidative burst in *S.* Paratyphi A and suggested that it is required for serum resistance, but the contribution of L O-ag was not assessed [20]. In the present study, we compared the contributions of L and VL O-ag to serum resistance and complement activation in *S.* Typhimurium, *S.* Typhi, and *S.* Paratyphi A by measuring serum

bactericidal activity, complement deposition, immunoglobulin binding, and anaphylatoxin release of wild-type and mutant *Salmonella* strains lacking WzzB, FepE, and/or the Vi capsular polysaccharide. Our novel results show that VL O-ag is the major mechanism of serum resistance in *S.* Paratyphi A, that duplication of the *rfbV* glycosyltransferase gene contributes to serum resistance in *S.* Paratyphi A by enhancing the quantity of O-ag production, that L O-ag is deficient in *S.* Paratyphi A due to a WzzB point mutation, and that O-ag length and composition significantly impact immunoglobulin binding and complement activation.

## Results

### Nontyphoidal and enteric fever *Salmonella* serovars differ in mechanisms of serum resistance and binding of complement and immunoglobulins

Serum resistance and binding of complement and immunoglobulins by wild-type *S.* Typhimurium, *S.* Typhi, and *S.* Paratyphi A and their mutant derivatives were compared. Isogenic *wzzB*, *fepE*, and *vexA* mutant derivatives were constructed to eliminate production of L O-ag, VL O-ag (in *S.* Typhimurium and *S.* Paratyphi A only), or Vi capsule (in *S.* Typhi only), respectively. Bacteria were incubated with 50% human serum at 37 °C for 60 min, diluted, and plated for enumeration of viable cells. All wild-type serovars were resistant to serum killing, but the contribution of L or VL O-ag to serum resistance was serovar-dependent (Fig 1A). Loss of WzzB-dependent L O-ag production increased serum sensitivity in *S.* Typhimurium and *S.* Typhi but not in *S.* Paratyphi A. Loss of FepE-dependent VL O-ag production did not significantly increase serum sensitivity in *S.* Typhimurium but rendered *S.* Paratyphi A highly susceptible to serum killing. Although *S.* Typhi does not produce VL O-ag because *fepE* is a pseudogene, loss of Vi capsule increased sensitivity to serum killing and had an additive effect when combined with loss of L O-ag. None of the three serovars were able to survive serum exposure in the absence of L and VL O-ag and the Vi capsule.

Deposition of complement component C3b is an essential step in complement-mediated serum killing [22]. Therefore, we used flow cytometry to determine the effect of O-ag length on the deposition of C3b on the bacterial surface. *S.* Typhimurium exhibits higher levels of C3b deposition in comparison to *S.* Typhi and *S.* Paratyphi A (Fig 1B and 1C). L and VL O-ag have only modest effects on the percent of *S.* Typhimurium with C3b deposition, even though the loss of both WzzB and FepE results in increased serum killing, indicating that serum resistance in *S.* Typhimurium does not result from avoidance of C3b deposition. Instead, serum resistance in *S.* Typhimurium appears to depend on the localization of C3b deposition, as a *wzzB* mutant, which lacks L O-ag, has less total C3b deposition yet is more susceptible to serum killing. Similarly, the loss of L O-ag in *S.* Typhi increases serum susceptibility but does not significantly increase C3b deposition. In contrast, loss of the Vi capsule significantly increases the percent of *S.* Typhi with C3b deposition, corroborating previous studies of its anti-opsonic function [32]. In *S.* Paratyphi A, VL O-ag was found to be the major determinant of C3b deposition as well as serum killing (Fig 1A). Loss of WzzB-dependent L O-ag production was observed to result in a small but significant decrease in C3b deposition in *S.* Paratyphi A.

Serum immunoglobulins bind antigen to catalyze complement activation, and bacteria have evolved mechanisms to inhibit antibody recognition and the alternative complement pathway [41–44]. We analyzed the effect of L and VL O-ag on the binding of IgG (Fig 1D) and IgM (Fig 1E) to *Salmonella*. All three serovars exhibited similar levels of IgG binding. In *S.* Typhimurium, loss of L and VL O-ag results in a small but significant decrease in IgG binding. The *S.* Typhi Vi capsule prevents IgG binding, but loss of L O-ag does not have a significant effect on IgG binding. Similarly, loss of VL O-ag in *S.* Paratyphi A increases IgG binding. The loss of L and VL O-ag reduces IgG binding compared to loss of VL O-ag alone, suggesting that IgG may bind to L O-ag in the absence of VL O-ag in *S.* Paratyphi A.

As with C3b deposition, higher levels of IgM binding were observed in *S.* Typhimurium compared to *S.* Typhi and *S.* Paratyphi A. IgM binding to *S.* Typhimurium is dependent on the presence of both L and VL O-ag (Fig 1E). Although some IgM binding of wildtype *S.* Typhi is observed, increased IgM binding is seen in the absence of Vi capsule, but not in strains lacking both Vi capsule and L O-ag, suggesting that the increased IgM binding is attributable to unmasking of L O-ag

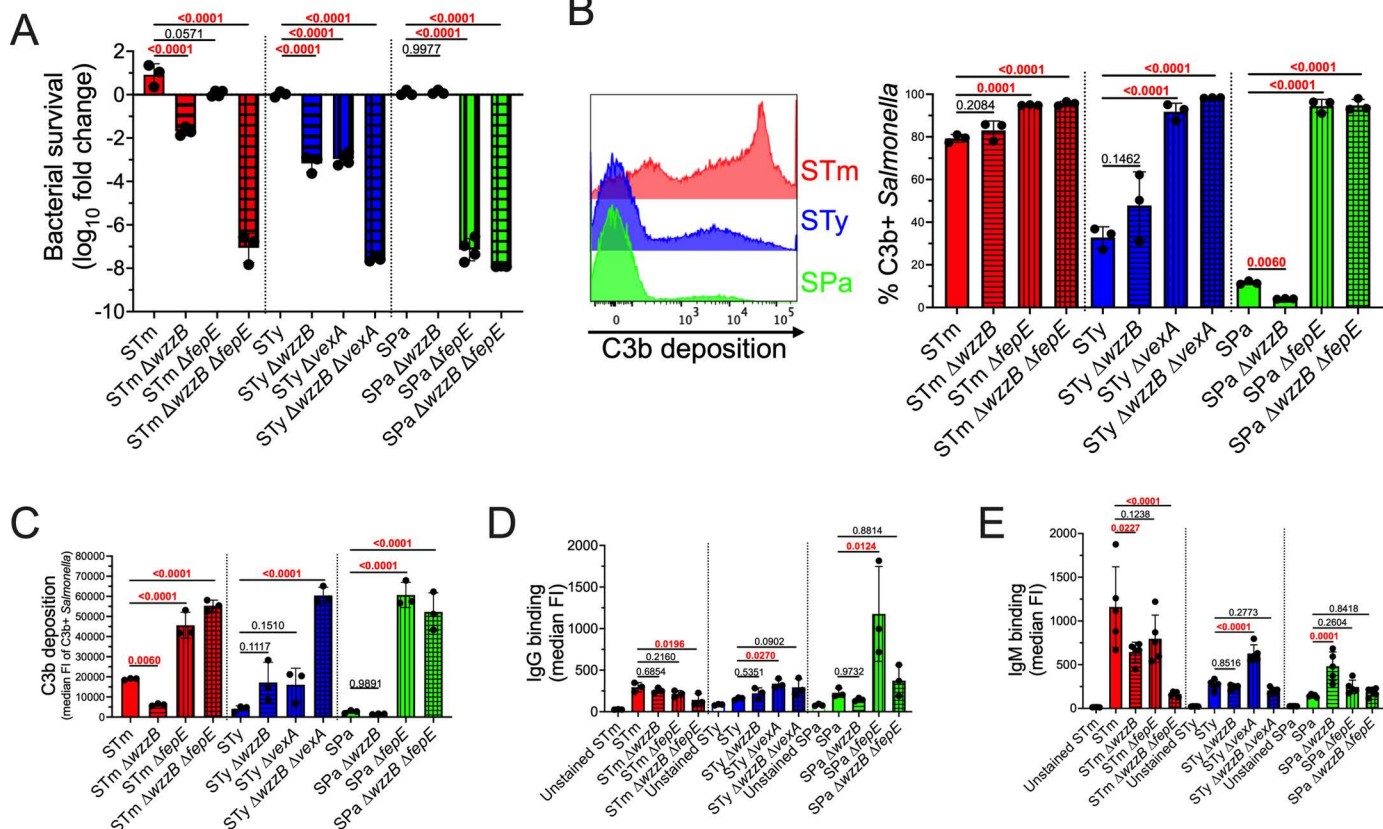

**Fig 1. The contribution of O-ag length and the Vi capsule to serum resistance is serovar dependent.** (A) Serum Resistance. Wild-type *S.* Typhimurium, *S.* Typhi, and *S.* Paratyphi A are resistant to complement-mediated killing. *S.* Typhimurium depends on both L and VL O-ag to resist complement killing, whereas serum resistance of *S.* Typhi is dependent on Vi capsule and L O-ag. *S.* Paratyphi A requires VL O-ag to resist serum killing, but L O-ag is dispensable. Complement bactericidal activity was assayed in 50% human pooled serum after 60 min incubation at 37 °C. An 8-log reduction in bacterial survival is the limit of detection for serum bactericidal assays. (B) and (C) C3b Deposition. Complement resistance mediated by the Vi capsule and by L and VL O-ag in *S.* Typhi and *S.* Paratyphi A is inversely proportional to C3b deposition. However, *S.* Typhimurium serum resistance is not associated with reduced C3b deposition. Representative flow cytometry histograms show increased C3b deposition on *S.* Typhimurium compared to *S.* Typhi and *S.* Paratyphi A. (D) IgG Binding. *S.* Typhimurium L and VL O-ag enhances IgG binding, whereas the *S.* Typhi Vi capsule prevents IgG binding. *S.* Paratyphi A VL O-ag prevents IgG binding. (E) IgM Binding. L and VL O-ag are the primary IgM antigenic binding determinants in *S.* Typhimurium. The *S.* Typhi Vi capsule prevents IgM binding. Deficient L O-ag in *S.* Paratyphi A enhances IgM binding. C3b and immunoglobulin binding were determined following 20 min incubation at 37 °C in 50% human pooled serum. Statistical analysis was performed by one-way ANOVA on 3–5 biological replicates, with P values in red indicating statistical significance with P < 0.05. Column bars represent means, with error bars showing standard deviation. STm, *S.* Typhimurium; STy, *S.* Typhi; SPa, *S.* Paratyphi A; FI, fluorescence intensity.

epitopes in the absence of Vi capsule. However, the loss of L O-ag increases IgM binding to *S.* Paratyphi A, and this is abrogated in the absence of both L and VL O-ag. Collectively, these results show that nontyphoidal *S.* Typhimurium exhibits higher levels of C3b and IgM binding than the enteric fever serovars *S.* Typhi and *S.* Paratyphi A, and the determinants of resistance to serum killing, complement deposition and immunoglobulin binding are serovar-dependent.

### WzzB-dependent L O-ag production is defective in *S.* Paratyphi A due to an arginine-98-to-cysteine mutation

As *S.* Paratyphi A is entirely dependent on FepE-mediated VL O-ag for serum resistance, and the loss of WzzB had no measurable impact, we compared O-ag production among *Salmonella* serovars by gel electrophoresis (Fig 2A). *S.* Typhimurium and *S.* Typhi exhibited an expected L O-ag profile with 16-to-35 O-ag repeating units. However, L O-ag

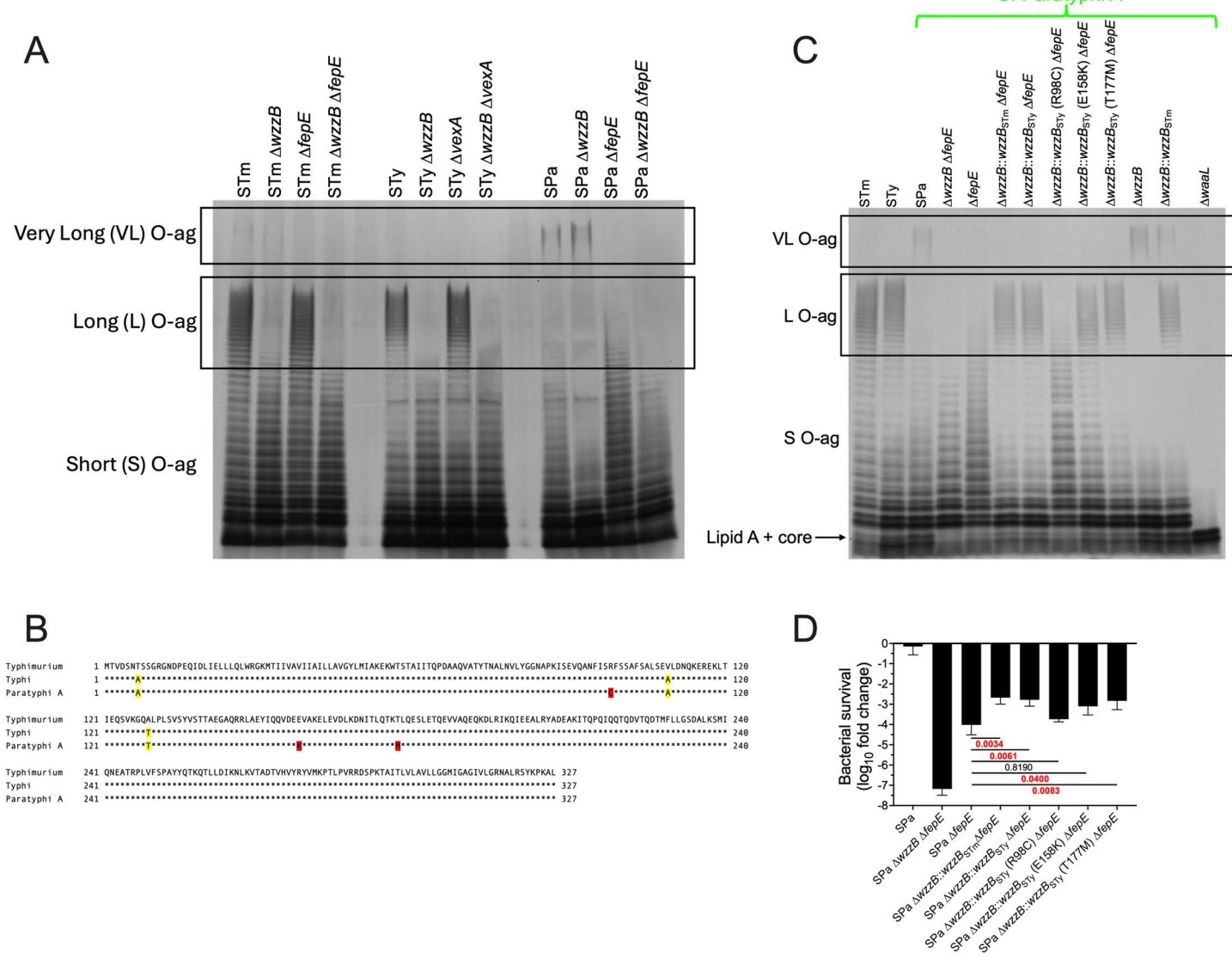

**Fig 2. *S.* Paratyphi A has deficient L O-ag production due to R98C mutation in WzzB.** (A) LPS extraction followed by gel electrophoresis and silver staining shows that *S.* Paratyphi A L O-ag production is deficient, whereas VL O-ag production is increased compared to *S.* Typhimurium. (B) Amino acid sequence of WzzB in *S.* Typhimurium, *S.* Typhi, and *S.* Paratyphi A. Mutations shared by *S.* Typhi and *S.* Paratyphi A in relation to *S.* Typhimurium are highlighted in yellow. Mutations unique to *S.* Paratyphi A WzzB are highlighted in red. (C) Allelic exchange of *S.* Paratyphi A *wzzB* with *S.* Typhimurium or *S.* Typhi *wzzB* restores L O-ag production by *S.* Paratyphi A. A single amino acid mutation in *S.* Paratyphi A WzzB from Arg to Cys at position 98 reduces WzzB activity. (D) Restoration of *S.* Paratyphi A L O-ag production by functional WzzB confers serum resistance in the absence of VL O-ag. Complement bactericidal activity was assayed in 25% pooled human serum after 60 min incubation at 37 °C. An 8-log reduction in bacterial survival is the limit of detection for serum bactericidal assays. Statistical analysis was performed by one-way ANOVA on 3 biological replicates, with P values in red indicating statistical significance as $P < 0.05$. Column bars represent the means, with error bars showing standard deviation. STm, Typhimurium; STy, Typhi; SPa, Paratyphi A; O-ag, O-antigen.

was not visible in extracts from *S.* Paratyphi A, which instead produces greater quantities of VL O-ag in comparison to *S.* Typhimurium. The elimination of FepE-dependent VL O-ag in *S.* Paratyphi A increased short O-ag production but failed to restore L O-ag production. These findings suggest the WzzB O-ag chain length regulator is hypofunctional in *S.* Paratyphi A, resulting in reduced quantities of L O-ag in *S.* Paratyphi A and accounting for the lack of an effect of a *wzzB* null

mutation on serum sensitivity in that serovar; this could also account for the dependence of *S.* Paratyphi A on VL O-ag for serum resistance (Fig 1A).

Protein sequence comparison showed that the *S.* Typhi and *S.* Paratyphi A WzzB proteins differ by three and six non-synonymous point mutations, respectively, compared to *S.* Typhimurium WzzB (Fig 2B). The three mutations in *S.* Typhi WzzB are shared with *S.* Paratyphi A, suggesting that these mutations are not responsible for reduced WzzB activity. Allelic exchange was performed to replace the *S.* Paratyphi A *wzzB* gene with *wzzB* from *S.* Typhimurium or *S.* Typhi, which restored L O-ag production in *S.* Paratyphi A (Fig 2C), indicating that one or more of the three unique non-synonymous mutations not found in *S.* Typhi WzzB is responsible for reduced activity of *S.* Paratyphi A WzzB. After replacement of each of these residues in *S.* Typhi WzzB, mutation of Arg98-to-Cys (R98C) was found to reduce the activity of *S.* Typhi WzzB and produced a L O-ag profile similar to that of *S.* Paratyphi A WzzB (Fig 2C). The Glu158-to-Lys or Thr177-to-Met mutations did not affect L O-ag production. The effect of these mutations on *S.* Paratyphi A serum resistance was also determined (Fig 2D). Restoration of *S.* Paratyphi A L O-ag production with *S.* Typhimurium or *S.* Typhi *wzzB* allelic exchange increased resistance to human serum when VL O-ag production was absent. However, the increase in serum resistance was abrogated by a WzzB R98C mutation, in association with a reduction in L O-ag production. Measurement of resistance to a lower (25%) concentration of human serum showed that the L O-ag produced by the hypofunctional *S.* Paratyphi A WzzB protein still provides some resistance to serum killing in comparison to *S.* Paratyphi A that lacks both WzzB and FepE (Fig 2D). Collectively, these data show that the WzzB R98C mutation in *S.* Paratyphi A reduces L O-ag production.

### Restoration of L O-ag production in *S.* Paratyphi A reduces VL O-ag production and affects complement deposition and immunoglobulin binding

L and VL O-ag vary in chain length but not in composition. The chain length modulators WzzB and FepE compete for the same substrates, but the mechanism of chain length modulation is poorly understood [45]. Since *S.* Paratyphi A lacks L O-ag but exhibits robust VL O-ag production, we measured O-ag production by densitometry to determine whether VL O-ag production is affected by WzzB activity. Restoration of L O-ag production in *S.* Paratyphi A by allelic exchange with *S.* Typhimurium *wzzB* decreased VL O-ag production (Fig 3A and 3B). Thus, the production of VL O-ag in *S.* Paratyphi A appears to be inversely proportional to WzzB activity.

Since restoration of L O-ag production by *wzzB* allelic exchange in *S.* Paratyphi A decreased VL O-ag production, we determined whether the restoration of L O-ag production also affected C3b deposition and immunoglobulin binding. *S.* Paratyphi A with restored L O-ag production was found to be more resistant to C3b deposition even though it produces less VL O-ag (Fig 3C and 3D). In contrast, replacement of *S.* Typhimurium *wzzB* with hypofunctional *wzzB* from *S.* Paratyphi A decreased L O-ag production and increased C3b deposition. In the absence of FepE-dependent VL O-ag production, no significant differences in C3b deposition were observed, indicating that the production of L O-ag is not sufficient to prevent C3b deposition. No differences were observed in IgG binding to *S.* Typhimurium expressing the hypofunctional WzzB from *S.* Paratyphi A (Fig 3E). However, restoration of L O-ag production in *S.* Paratyphi A decreased IgG binding (Fig 3E) while increasing IgM binding (Fig 3F).

### Hypofunctional WzzB and paratose-containing O-ag in *S.* Paratyphi A reduce IgM binding and complement activation

C3b deposition on *S.* Paratyphi A remains low in comparison to *S.* Typhimurium and *S.* Typhi despite a reduction of VL O-ag in strains with restored WzzB-dependent L O-ag production (Fig 3B and 3D). Therefore, we analyzed additional possible factors contributing to low C3b deposition in this *Salmonella* serovar. The repeating O-ag of *S.* Paratyphi A contains the dideoxyhexose paratose instead of tyvelose, which is present in *S.* Typhi, due to pseudogenization of *rfbE,* encoding a CDP-tyvelose epimerase that converts CDP-paratose to CDP-tyvelose. Previous studies have shown that the composition

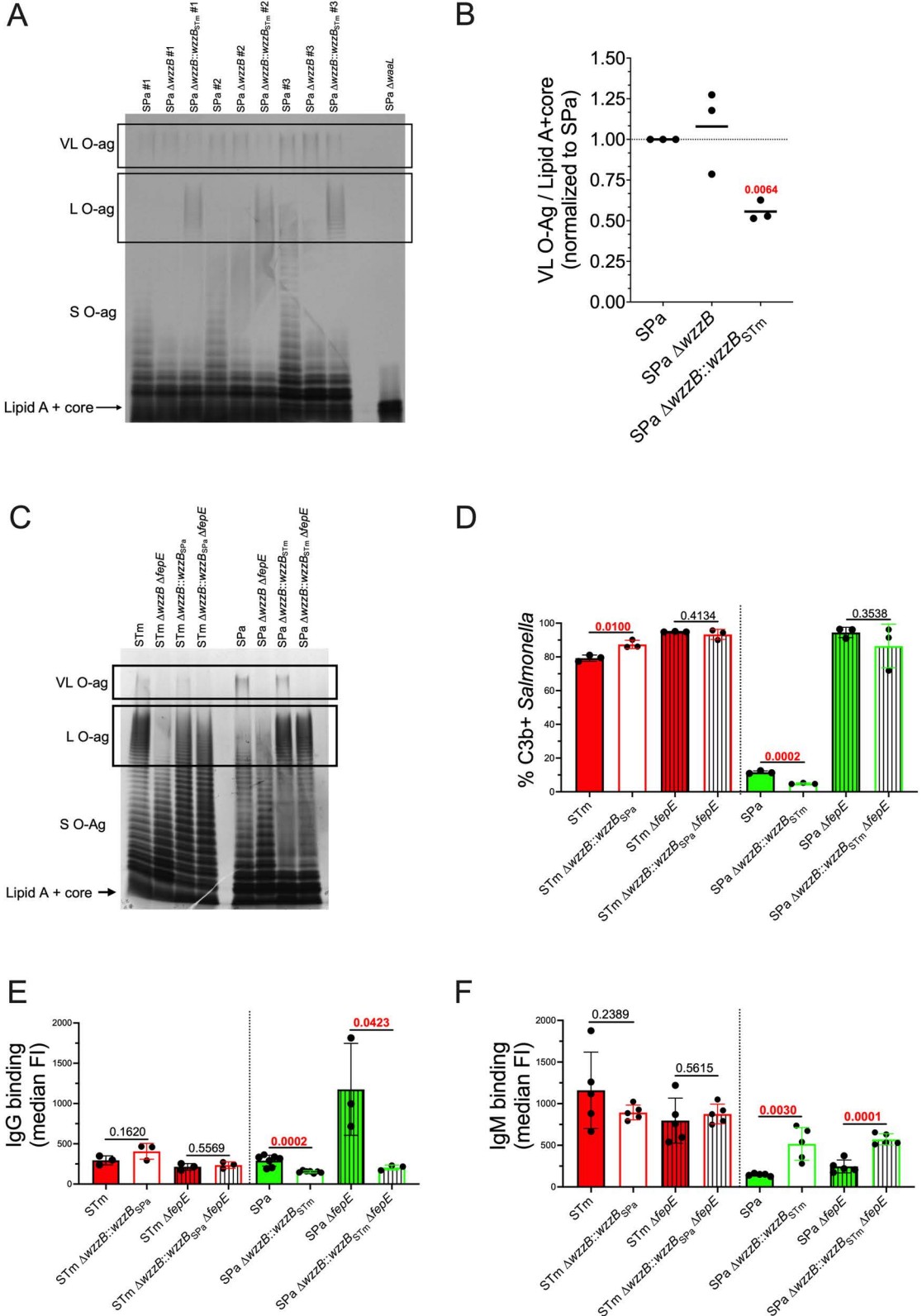

**Fig 3. Restoration of *S.* Paratyphi A L O-ag reduces VL O-ag production and impacts IgM binding.** (A) Three independent *S.* Paratyphi A LPS extractions followed by gel electrophoresis and silver staining were performed to show restoration of L O-ag and reduction of VL O-ag. (B) Densitometry

analysis. Statistical analysis was performed by one sample t test with a hypothetical value of 1.0 from 3 independent LPS extractions, with P values in red indicating statistical significance with P < 0.05. Line in the scatter dot plot represents the mean. (C) LPS extraction followed by gel electrophoresis and silver staining shows that *wzzB* allelic exchange shifts the modal L O-ag and restores L O-ag production in *S.* Typhimurium and *S.* Paratyphi A, respectively. (D) *wzzB* allelic exchange has small but significant effects on C3b deposition in *S.* Typhimurium and *S.* Paratyphi A. (E) Restoration of *S.* Paratyphi A L O-ag production reduces IgG binding but (F) increases IgM binding. C3b and immunoglobulin binding were determined following 20 min incubation at 37 °C in 50% pooled NHS. Statistical analysis was performed by t test on 3–7 biological replicates, with P values in red indicating statistical significance with P < 0.05. Some of the C3b deposition, IgG and IgM median FI values for STm, STmΔ*fepE*, SPa, and SPaΔ*fepE* are from the same experiment shown in Fig 1B-D but compared here to the corresponding *wzzB* allelic exchange construct. Column bars represent the means, with error bars showing standard deviation. STm, Typhimurium; STy, Typhi; SPa, Paratyphi A; FI, fluorescence intensity.

of O-ag can impact C3b deposition [46–48]. Under our experimental conditions, repairing *S.* Paratyphi A *rfbE* by allelic exchange with functional *S.* Typhi *rfbE* did not affect the percentage of bacteria with C3b deposition (Fig 4A) but lower quantities of C3b were detected on bacteria with C3b deposition (Fig 4B). Conversely, restoring L O-ag production in *S.* Paratyphi A by allelic exchange with functional *wzzB* from *S.* Typhimurium decreased the overall percent of bacteria with C3b deposition (Fig 4A) but did not affect the quantity of C3b deposition on these cells (Fig 4B). These differences in C3b deposition were not observed in *S.* Paratyphi A that produced both long tyvelose-containing O-ag (Fig 4A and 4B). Similarly, IgG binding by *S.* Paratyphi A was reduced when L O-ag production was restored, with a smaller reduction observed in a strain expressing a functional *rfbE* gene, but no significant differences in IgG binding were measured when both *wzzB* and *rfbE* were repaired (Fig 4C). In contrast, the restoration of either L O-ag or tyvelose O-ag production in *S.* Paratyphi A significantly increased IgM binding (Fig 4D), which increased further when both *wzzB* and *rfbE* were repaired. The increase in IgM binding as a result of *rfbE* allelic exchange was not attributable to restored L O-ag production, suggesting that *S.* Paratyphi A WzzB remains hypofunctional when the O-ag contains tyvelose (Fig 4E). These changes in complement deposition and immunoglobulin binding did not affect serum resistance in *S.* Paratyphi A (Fig 4F). Collectively, these observations suggest that reduced L O-ag production and modified O-ag composition contribute to the reduced IgM binding of *S.* Paratyphi A.

Complement activation leads to the production of the soluble anaphylatoxins C3a and C5a. We measured the contribution of O-ag length and composition on anaphylatoxin production in human serum following exposure to nontyphoidal and enteric fever *Salmonella* serovars. Differences in complement C3a production among serovars were subtle (Fig 5A), but *S.* Typhimurium induced greater C5a production than the enteric fever serovars (Fig 5B). L O-ag production in *S.* Typhimurium was required for increased C5a production, while VL O-ag was dispensable (Fig 5B). Both L O-ag and the Vi capsule contributed to reduced C3a production following exposure of serum to *S.* Typhi, but no increase in C5a production was observed in mutant strains. In *S.* Paratyphi A, VL O-ag prevented C3a and C5a production, in agreement with our earlier results indicating that VL O-ag is the primary determinant of complement resistance in *S.* Paratyphi A. Restoration of L O-ag and tyvelose O-ag production in *S.* Paratyphi A, which increased IgM binding, also increased C3a and C5a production. These observations suggest that reduced L O-ag production and modified O-ag composition contribute to decrease anaphylatoxin elicitation by *S.* Paratyphi A.

### O-ag modifications by glycosyltransferases in *S.* Typhimurium and *S.* Paratyphi A modulate complement activation and IgM binding

O-ag length and composition are important determinants of IgM binding to *S.* Paratyphi A (Figs 4 and 5). However, even with increased IgM binding (Fig 4D) and complement activation (Fig 5A and 5B), *S.* Paratyphi A exhibits reduced complement deposition compared to *S.* Typhimurium though it lacks the *S.* Typhi Vi capsule. We sought to determine whether *S.* Typhimurium and *S.* Paratyphi A O-ag-modifying glycosyltransferases influence complement activation and deposition, IgM binding, and serum resistance, as these modifications have been previously shown to impact antibody binding and serum resistance in other serovars [49]. Three O-ag-modifying glycosyltransferases have been identified in *S.* Paratyphi

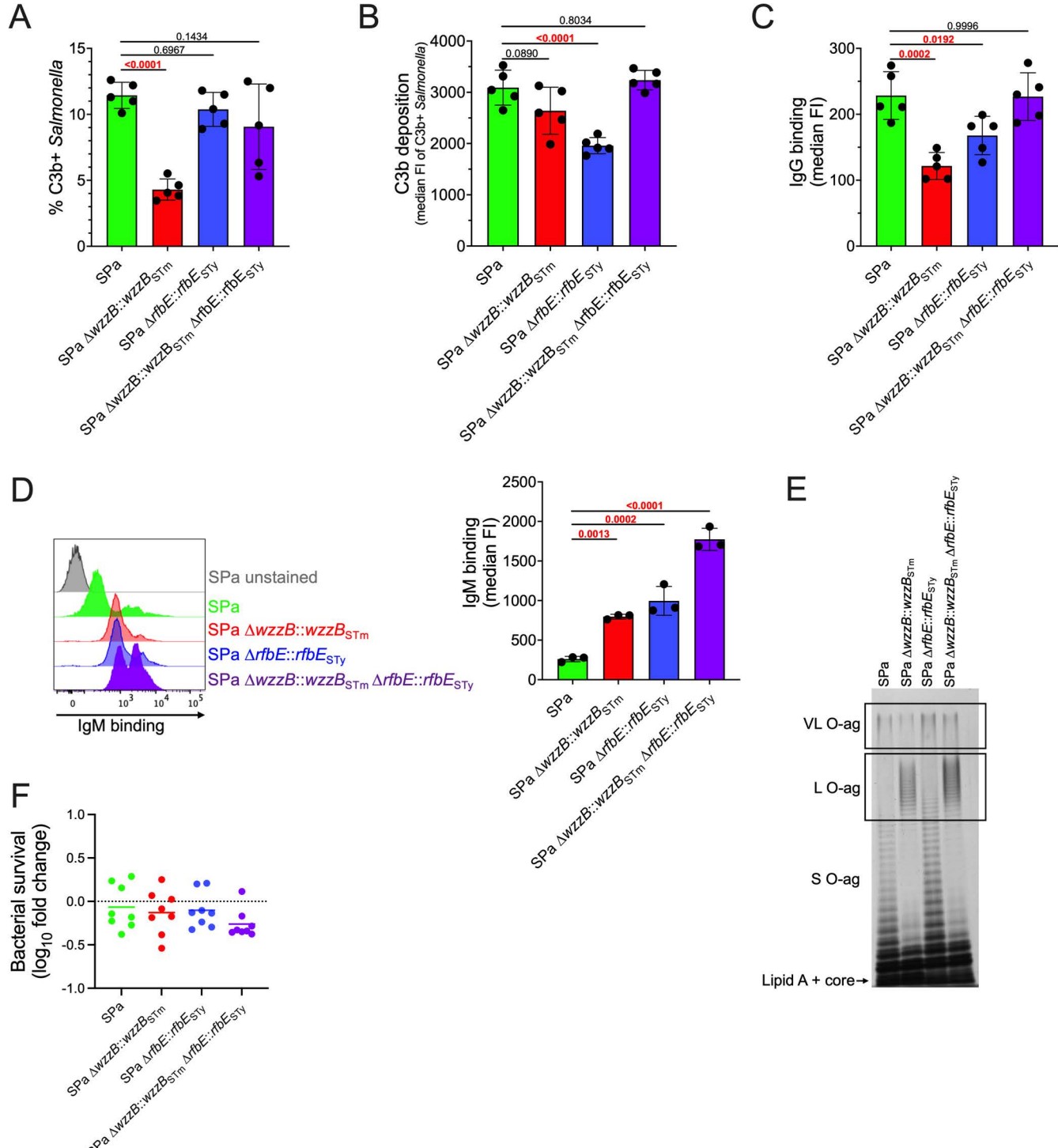

**Fig 4. Reduction of L O-ag production and antigenic shift to O2-antigen in *S.* Paratyphi A decreases IgM binding. (**A) and (B) Restored *S.* Paratyphi A L O-ag production further reduces complement C3b deposition, but the presence of O9-antigen abrogates this effect. (C) IgG binding is reduced following restoration of L O-ag or O9-antigen production in *S.* Paratyphi A. (D) Restoration of L O-ag and O9-antigen production increases IgM binding of *S.* Paratyphi A, but (F) does not increase sensitivity to serum killing. Representative flow cytometry histogram in (D) displays increased IgM binding to *S.* Paratyphi A due to restoration of L O-ag and O9-antigen. (E) Representative LPS extraction showing that a shift to O9-antigen does not restore L O-ag production in *S.* Paratyphi A. C3b and immunoglobulin binding were determined following 20 min incubation at 37 °C in 50% human pooled serum,

while serum resistance was determined following 60 min incubation. An 8-log reduction in bacterial survival is the limit of detection for serum bactericidal assays. Statistical analysis was performed by one-way ANOVA on 3–5 biological replicates, with P values in red indicating statistical significance with P < 0.05. Column bars represent the means, with error bars showing standard deviation. STm, *S*. Typhimurium; STy, *S*. Typhi; SPa, *S*. Paratyphi A; FI, fluorescence intensity.

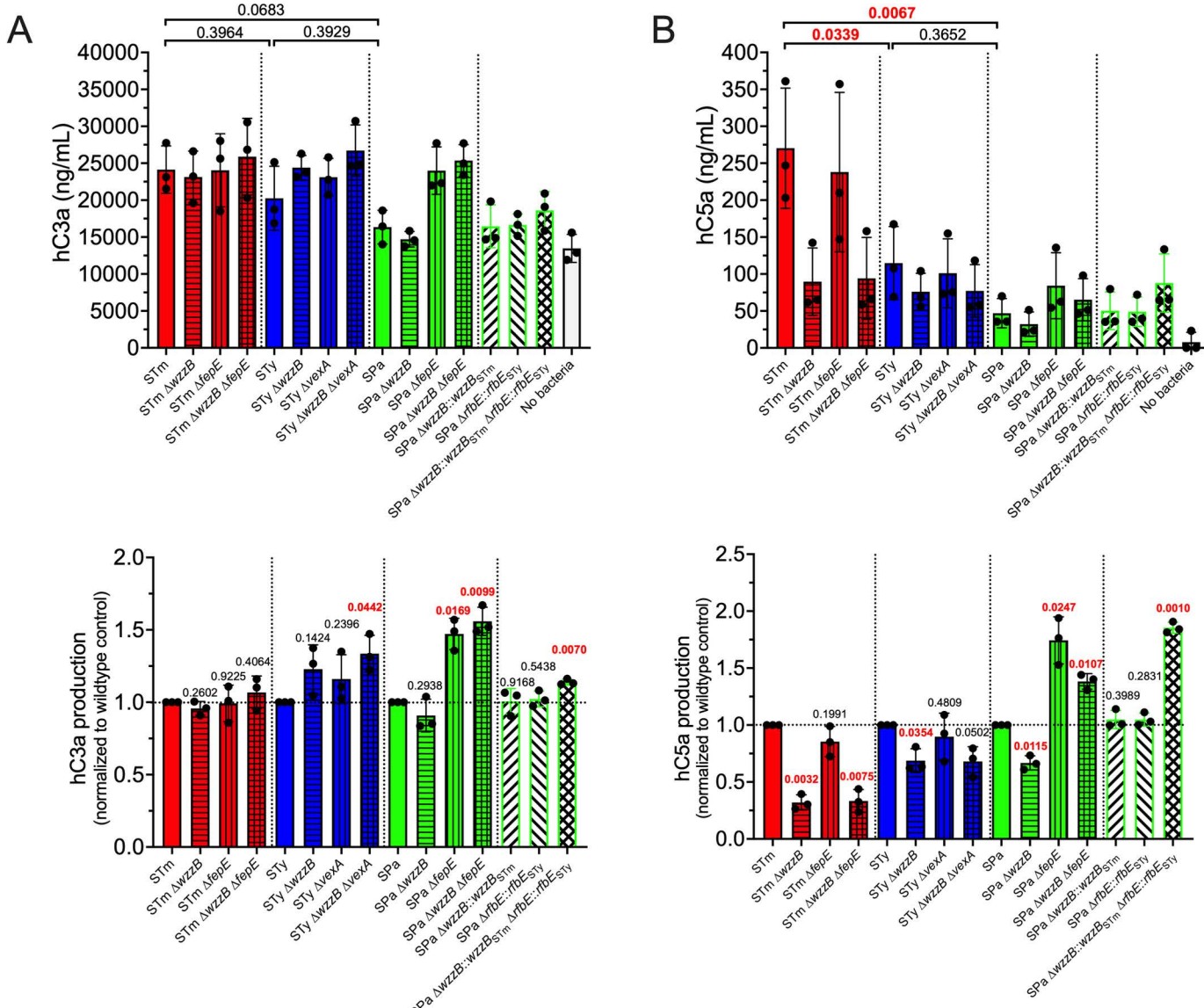

**Fig 5. O-ag length and composition impacts complement activation by *Salmonella* nontyphoidal and enteric fever serovars.** (A) Restoration of *S*. Paratyphi A L O-ag and O9-antigen increases human C3a and (B) C5a production following incubation with human serum for 60 min at 37°C. The *S*. Typhi Vi capsule prevents C3a production but does not impact C5a release. L O-ag in *S*. Typhimurium is the primary determinant of C5a release. C3a and C5a release were measured by enzyme-linked immunosorbent assay (ELISA). For statistical comparison, individual replicates are normalized to the corresponding wild-type serovar and statistical analysis performed by one sample t test with a hypothetical value of 1.0, with P values in red indicating statistical significance with P < 0.05. Column bars represent the means, with error bars showing standard deviation. STm, *S*. Typhimurium; STy, *S*. Typhi; SPa, *S*. Paratyphi A.

A: the OafB acetyltransferase (encoded by *SPA0467,* also known as family II *gtrC* or *F2gtrC*) acetylates the 2- and 3-hydroxyl groups of rhamnose of the O-ag repeating unit, and the F1gtrC (*SPA2387*) and F3gtrC (*SPA2169*) glycosyltransferases attach a glucosyl group to the 6- or 4-hydroxyl group of galactose in the O-ag repeating unit, respectively [50–54]. Deletion of OafB and F3gtrC increased C5a release and IgM binding in *S.* Paratyphi A, but deletion of F1gtrC did not (Fig 6A and 6B). Deletion of the acetyltransferase and glycosyltransferases did not impact C3b deposition or serum resistance (Fig 6C, 6D, and 6E). Deletion of the abequose acetyltransferase OafA in *S.* Typhimurium did not impact C5a release but increased IgM binding (Fig 6F and 6G) [55,56]. Deletion of the glycosyltransferases F3gtrC or the putative F4gtrC, with unknown O-ag modifying function, also failed to impact C5a anaphylatoxin release by *S.* Typhimurium (Fig 6F). However, F4gtrC deletion increased IgM binding (Fig 6G). Similar to *S.* Paratyphi A, deletion of these O-ag modifying enzymes did not impact C3b deposition or serum resistance (Fig 6H, 6I, and 6J). Collectively, these results suggest that the effect of O-ag modifications by acetyltransferases and glycosyltransferases on complement activation and antibody binding are serovar dependent but are not determinants of C3b deposition or serum resistance in *S.* Paratyphi A and *S.* Typhimurium.

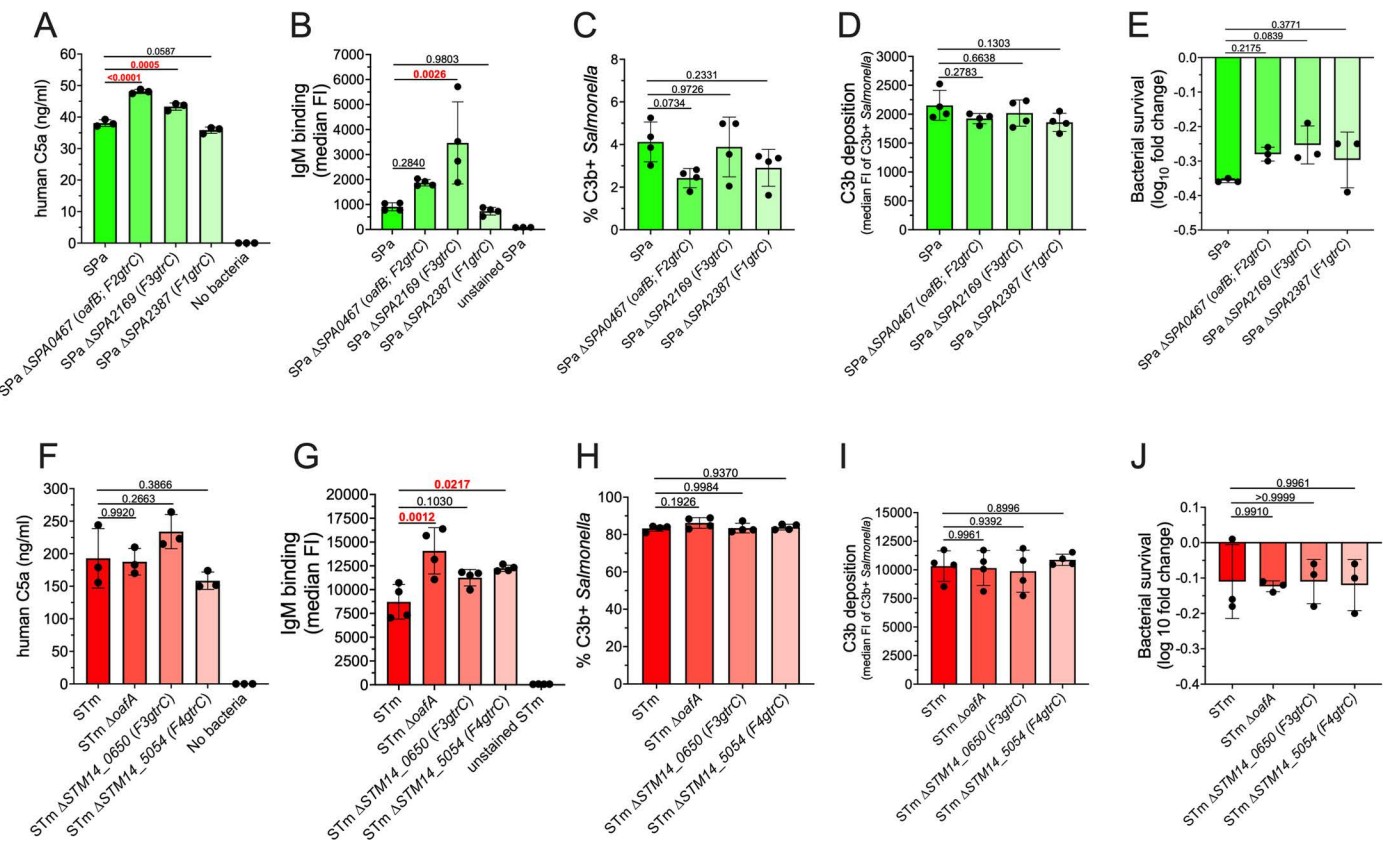

**Fig 6. Effect of O-ag modifying glycosyltransferases on complement activation and *Salmonella* interactions with human serum.** (A) O-ag modifications by glycosyltransferases in *S.* Paratyphi A decrease complement activation as measured by C5a release and (B) IgM binding. (C) and (D) O-ag modifications by glycosyltransferases in *S.* Paratyphi A do not impact complement deposition or (E) serum resistance. (F) O-ag modifications by glycosyltransferases in *S.* Typhimurium do not impact complement activation, (H) and (I) C3b deposition, or (J) serum resistance, but (G) reduce IgM binding. An 8-log reduction in bacterial survival is the limit of detection for serum bactericidal assays. Statistical analysis was performed by one-way ANOVA on 3–4 biological replicates, with P values in red indicating statistical significance with $P < 0.05$. Column bars represent the means, with error bars displaying standard deviation. STm, *S.* Typhimurium; STy, *S.* Typhi; SPa, *S.* Paratyphi A.

**Multiple gene copies of *rfbV* in *S.* Paratyphi A increase VL O-ag production, prevent C3b deposition, and enhance serum resistance**

*S.* Paratyphi A produces greater quantities of VL O-ag than *S.* Typhimurium (Fig 2A). We hypothesized that increased VL O-ag production in *S.* Paratyphi A prevents C3b deposition, while reduced VL O-ag production in *S.* Typhimurium permits C3b deposition. First, we sought to determine an explanation for increased VL O-ag levels in *S.* Paratyphi A. The O-ag locus of *S.* Paratyphi A strain ATCC 9150 contains three gene copies of *rfbV* (also known as *wbaV*), which encodes an enzyme that links paratose to mannose of the O-ag repeating unit before it is flipped to the periplasmic face by RfbX (also known as Wzx); *S.* Typhimurium and *S.* Typhi contain only a single copy of *rfbV* [57–60]. The extra copies of *rfbV* in *S.* Paratyphi A are preceded by a hypothetical open reading frame (ORF) composed of the C-terminus of RfbX and the N-terminus of RfbU, which attaches rhamnose to mannose in the O-ag repeating unit (Fig 7A). To determine if increased *rfbV* gene copy number enhances VL O-ag to prevent C3b deposition and increase serum resistance, we constructed a *S.* Paratyphi A strain that contains only one copy of *rfbV* (Fig 7A). The constructed *S.* Paratyphi A strain with a single copy of *rfbV* was more susceptible to serum killing compared to the wild-type *S.* Paratyphi A strain with three copies of *rfbV* (Fig 7B). Serum resistance in the constructed *S.* Paratyphi A strain with a single copy of *rfbV* was fully restored by constitutively expressing either *rfbV* or *rfbV* preceded by the hypothetical *rfbU-rfbX* fusion ORF, as found in wild-type *S.* Paratyphi A (Fig 7B). Interestingly, constitutive expression of only the hypothetical *rfbU-rfbX* fusion ORF also restored some serum resistance, suggesting that the putative fusion protein retains enzymatic activity affecting O-ag production (Fig 7B). Serum killing was proportional to C3b deposition (Fig 7C), and IgM binding was increased in the *S.* Paratyphi A strain carrying a single copy of *rfbV* (Fig 7D). O-ag staining and densitometry analysis showed that *S.* Paratyphi A with multiple *rfbV* copies produced more VL O-ag than the constructed strain with a single *rfbV* copy (Fig 7E and 7F). Enhanced VL O-ag production in *S.* Paratyphi A with a single *rfbV* copy was restored with constitutive expression of either *rfbV* or *rfbV* preceded by the hypothetical *rfbU-rfbX* fusion ORF, and more modestly increased by expression of only the hypothetical *rfbU-rfbX* fusion ORF alone (Figure 7E and 7F). Collectively, these results suggest that *rfbV* gene duplication in *S.* Paratyphi A increases VL O-ag production to prevent complement deposition and IgM binding, and to enhance serum resistance.

We reasoned that increasing *rfbV* production in *S.* Typhimurium might also increase VL O-ag production and prevent C3b deposition. However, constitutively expressing *rfbV* in Typhimurium did not increase VL O-ag production nor prevent C3b deposition, in contrast to *S.* Paratyphi A (S1 Fig). A previous study suggested that FepE levels are higher in *S.* Paratyphi A compared to *S.* Typhimurium [61]. To determine whether increased *fepE* expression can increase VL O-ag production, we constitutively expressed *fepE* in *S.* Typhimurium, which resulted in increased VL O-ag production (Fig 7G) and decreased C3b deposition (Fig 7H). L O-ag length was found to inhibit the anti-opsonic effect of increased VL O-ag production, since *S.* Typhimurium with L O-ag was more susceptible to C3b deposition compared to *S.* Typhimurium with shorter L O-ag following allelic exchange with the hypofunctional *wzzB* from *S.* Paratyphi A or lacking L O-ag altogether (Fig 7H). Similarly, L O-ag increased IgM binding (Fig 7I). These findings suggest that high levels of VL O-ag are required for complement evasion, that L O-ag promotes complement deposition and IgM binding in *S.* Typhimurium, and that *rfbV* gene duplication in *S.* Paratyphi A is required to inhibit complement deposition and IgM binding.

## Discussion

*Salmonella enterica* lipopolysaccharide, with its repeating O-ag units, functions as a physicochemical barrier that modulates host recognition, phagocyte ingestion, susceptibility to bactericidal mechanisms, and stimulation of innate immunity via the LPS-binding protein—CD14—TLR4—MD-2 complex [28,29,33,62]. Important differences in this outermost layer of the cell envelope of nontyphoidal and enteric fever *Salmonella* serovars result in critical differences in the respective responses to these pathogens that are initiated when they interact with the host. In this study, we have compared the roles of L and VL O-ag and the Vi capsule in conferring resistance to serum killing, complement and immunoglobulin binding, and complement activation in nontyphoidal *S.* Typhimurium and the enteric fever serovars *S.* Typhi and *S.* Paratyphi A (Fig

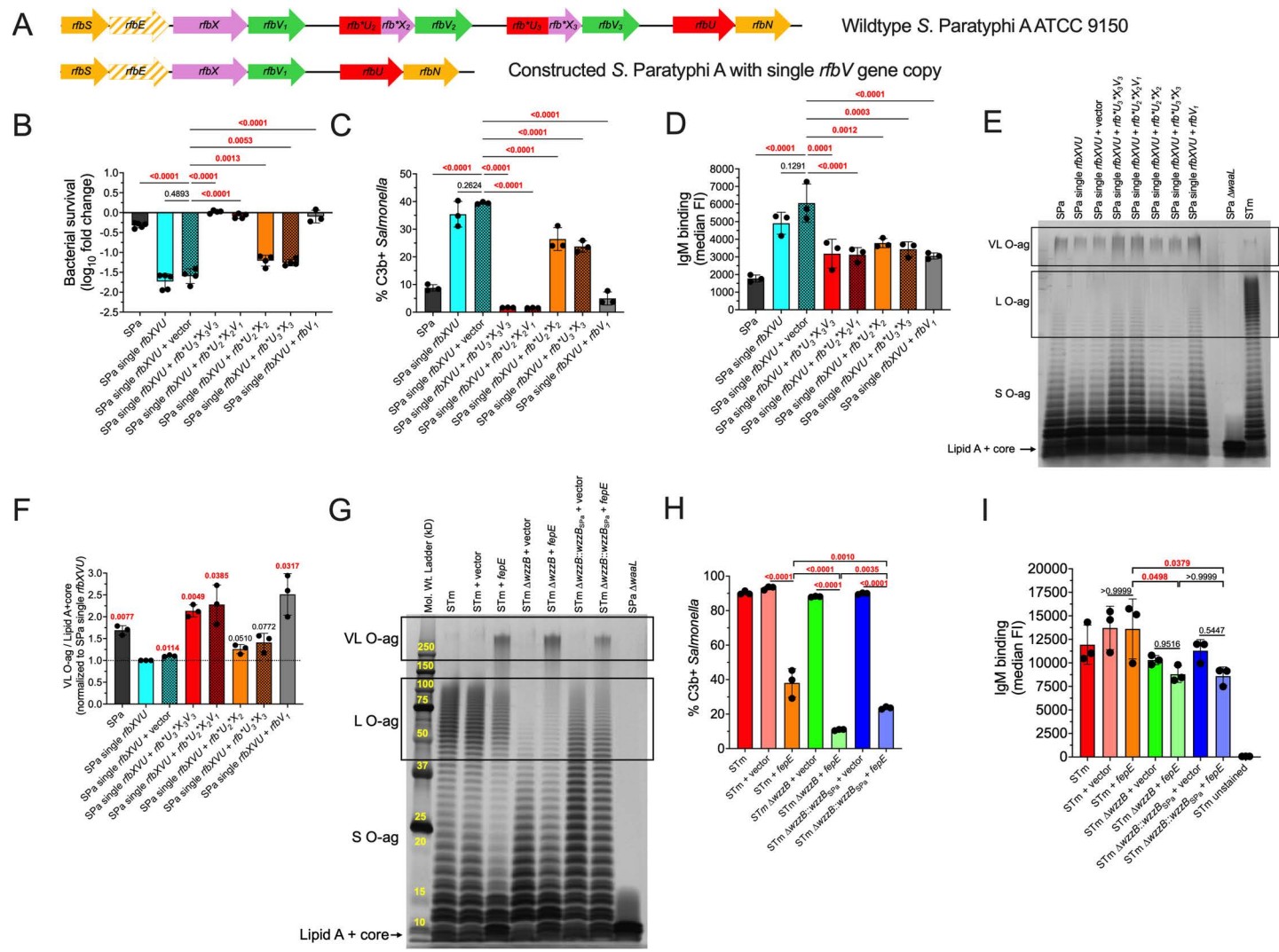

**Fig 7. *rfbV* gene triplication in S. Paratyphi A increases VL O-ag production and reduces C3b deposition.** (A) Gene diagram displaying triplicated *rfbV* gene in wild-type *S.* Paratyphi A ATCC 9150 and the constructed strain with a single *rfbV* gene copy. The nonfunctional *rfbE* gene is displayed by a dashed arrow, and partial genes are denoted by an asterisk. (B) Serum resistance. *S.* Paratyphi A with a single gene copy of *rfbV* is less resistant to serum killing compared to wild-type *S.* Paratyphi A that contains three *rfbV* gene copies. *S.* Paratyphi A complement resistance is restored by complementation with *rfbV* alone or *rfbV* preceded by the putative fusion protein encoded by incomplete copies of *rfbU* and *rfbX,* which are denoted by an asterisk. Complementation with the putative fusion protein composed of Rfb*U*X also confers partial serum resistance. (C) C3b Deposition. *S.* Paratyphi A with single gene copy of *rfbV* is more susceptible to C3b deposition. Serum resistance is directly proportional to C3b deposition. (D) IgM Binding. Multiple *rfbV* gene copies reduce IgM binding. Reduced IgM binding is also observed when the putative Rfb*U*X fusion protein is expressed. (E) Extracted LPS subjected to gel electrophoresis followed by silver staining showing decreased VL O-ag production in *S.* Paratyphi A with a single *rfbV* gene copy compared to wild-type *S.* Paratyphi A ATCC 9150 with three *rfbV* gene copies. (F) Densitometry analysis from three independent LPS extractions. Statistical analysis was performed by one sample t test with a hypothetical value of 1.0, with P values in red indicating statistical significance as P<0.05. Column bars represent the mean with error bars showing standard deviation. (G) LPS extraction followed by silver staining showing increased VL O-ag production in *S.* Typhimurium that constitutively expresses *fepE*. (H) C3b Binding. Increased VL O-ag production in *S.* Typhimurium decreases C3b binding, whereas L O-ag promotes C3b binding. (I) IgM Binding. Increase in VL O-ag does not affect IgM binding to Typhimurium, whereas L O-ag increases IgM binding to Typhimurium. An 8-log reduction in bacterial survival is the limit of detection for serum bactericidal assays. Statistical analysis (except for Fig 7F) was performed by one-way ANOVA on 3–5 biological replicates, with P values in red indicating statistical significance with P<0.05. Column bars represent the means, with error bars showing standard deviation. STm, *S.* Typhimurium; STy, *S.* Typhi; SPa, *S.* Paratyphi A.

1). In agreement with previous studies, we found that L O-ag confers resistance to serum killing in *S.* Typhimurium and *S.* Typhi [30,31,39,63]. In contrast, WzzB-dependent L O-ag is not required for serum resistance of *S.* Paratyphi A when VL O-ag is present. LPS extractions suggested that *S.* Paratyphi A has relatively low levels of L O-ag, and we show that this results from an R98C mutation that reduces WzzB activity (Fig 2). Hypofunctional WzzB renders *S.* Paratyphi A reliant on FepE-dependent VL O-ag for serum resistance, which is in sharp distinction to *S.* Typhimurium, which does not depend on VL O-ag for serum resistance, as previously shown [29,33,39]. *S.* Typhi contains a *fepE* pseudogene and therefore does not produce VL O-ag, but its acquisition of the SPI-7 encoded Vi capsule provides an analogous structure that confers serum resistance, in agreement with previous observations [20,64]. Thus, nontyphoidal and enteric fever *Salmonella* serovars have evolved distinct mechanisms to resist serum killing.

*S.* Typhimurium binds the complement component C3b more avidly than *S.* Typhi or *S.* Paratyphi A (Fig 1B). The *S.* Typhi Vi capsule has been suggested to resist C3b deposition because it lacks free hydroxyl groups for nucleophilic attack on the susceptible complement C3b thioester bond [25,32,65]. O-ag repeating units containing free hydroxyl groups have been suggested to covalently attach to C3b by this mechanism [32]. However, the O-ag repeating units in *S.* Typhimurium, *S.* Typhi, and *S.* Paratyphi A differ only by the stereoisomers abequose, tyvelose, and paratose, respectively, yet *S.* Paratyphi A, which lacks the Vi capsule, binds less C3b than either *S.* Typhimurium or *S.* Typhi (Fig 1B and 1C). This suggests that additional factors modulate C3b deposition on the *Salmonella* surface.

The classical pathway of complement activation relies on antigen recognition by immunoglobulins to initiate complement deposition [22]. Human serum contains IgG and IgM that are able to bind all three *Salmonella enterica* serovars tested. The loss of O-ag in *S.* Typhimurium leads to a decrease in IgG and IgM binding, suggesting that O-ag contains a major epitope for immunoglobulin recognition (Fig 1D and 1E). The *S.* Typhi Vi capsule shields IgG and IgM O-ag epitope recognition. In *S.* Paratyphi A, VL O-ag plays a role analogous to that of the *S.* Typhi Vi capsule in preventing IgG recognition of L O-ag (Fig 1D). However, IgM epitopes on *S.* Paratyphi A appear to have a more complex antibody-antigen relationship, in which L O-ag and the loss of the O9 tyvelose antigen play a critical role. The paratose O2-antigen contributes to the decrease in IgM binding, since repairing the *S.* Paratyphi A *rfbE* pseudogene to restore production of the tyvelose O9-antigen significantly increases IgM binding (Fig 4), in agreement with Hiyoshi et al. [20]. We have also determined that restoration of *S.* Paratyphi A L O-ag production increases IgM binding, which is further increased when combined with restored O9-antigen production. Unexpectedly, increased binding of IgM, which is a potent complement activator, does not enhance complement C3b deposition on *S.* Paratyphi A but increases anaphylatoxin production (Fig 5). Paradoxically, either deletion of the hypofunctional *S.* Paratyphi A WzzB or restoration of fully functional WzzB-dependent L O-ag increased IgM binding (Figs 1E and 3F). We hypothesize that deletion of the *S.* Paratyphi A hypofunctional WzzB increases the accessibility of IgM epitopes closer to the bacterial surface, while restoration of L O-ag provides different IgM epitopes that are more accessible for IgM crosslinking than VL O-ag. Future studies may examine the physiochemical properties of L and VL O-ag that modulate immunoglobulin C3b binding to *S.* Paratyphi A. We did not observe a clear correlation between antibody binding and complement activation, suggesting a contribution from additional mechanisms of activation (e.g., the alternative and mannose-binding lectin pathways). Biophysical studies may be required to determine how O-ag length affects antibody binding and initiation of complement activation. We hypothesize that O-ag length may affect the multivalent antibody binding required for complement activation. In addition, studies are underway to determine how complement activation and opsonization affect interaction of host cells with nontyphoidal and enteric fever *S. enterica* serovars.

The present study clarifies and expands upon earlier observations regarding the role of LPS in *S.* Paratyphi A innate immune evasion. Mylona et al. suggested that increased production of *S.* Paratyphi A VL O-ag results from increased *fepE* expression relative to *S.* Typhimurium, and that this inhibits inflammasome activation and pyroptotic cell death [61]. We propose that increased VL O-ag production is also important for *S.* Paratyphi A immune evasion because it prevents binding of C3b and immunoglobulin and thereby prevents the recruitment of inflammatory cells by complement-derived

anaphylatoxins. We have further shown that the hypofunctional WzzB of *S.* Paratyphi A is responsible for increased VL O-ag production, most likely by diverting substrate to FepE (Fig 3A and 3B). Liu et al. have suggested that the main contributing factor to low levels of L O-ag in *S.* Paratyphi A is the inefficient attachment of paratose to the O-ag repeating unit by the glycosyltransferase RfbV, also known as WbaV [58]. However, one would expect low levels of both L and VL O-ag if RfbV were inefficient, but *S.* Paratyphi A in fact produces robust levels of VL O-ag. Based on our new findings, we propose that low levels of L O-ag production in *S.* Paratyphi A primarily result from hypofunctional WzzB. Multiple *rfbV* gene copies in *S.* Paratyphi A are required to produce enhanced levels of VL O-ag to evade host immune defenses. The importance of multiple *rfbV* gene copies is supported by recent findings showing that *S.* Paratyphi A clinical isolates contain more than two *rfbV* copies, with most preceded by a putative partial *rfbU* and *rfbX* gene fusion [66]. Our results also suggest that the putative RfbU-RfbX fusion protein retains enzymatic activity, since its expression appears sufficient to provide some resistance to serum killing, complement deposition, and decreased IgM binding (Fig 7). We speculate that the putative RfbU-RfbX fusion protein is needed in the presence of multiple *rfbV* copies in *S.* Paratyphi A to allow efficient production of the repeating O-ag unit. Although we were able to restore VL O-ag production by constitutive expression of the *rfbV* gene in *S.* Paratyphi A (Fig 7E), most plasmid constructs containing *rfbV* alone were found to contain mutations within the coding sequence. This suggests that increased RfbV enzymatic activity is not tolerated by *S.* Paratyphi A unless the putative RfbU-RfbX fusion protein is also present. The *S.* Paratyphi A constitutive *rfbV* construct used in this study contained a G nucleotide deletion in the ribosome binding site, which likely lowered RfbV protein to tolerable levels in the absence of the putative RfbU-RfbX fusion protein.

A limitation of the present study is the use of unfractionated serum, which may contain a mixture of different antibody types with different affinities for epitopes on the bacterial surface, as well as the different affinities of conjugated antibodies used to detect antibody binding. Therefore, relative binding of antibodies may not be indicative of absolute antibody binding that would require conjugated monoclonal antibodies and standardization. The present study shows that both L O-ag and Vi capsule independently contribute to *S.* Typhi resistance to serum killing, but the importance of capsule retention in *S.* Typhi and its role in serum resistance is uncertain. In *Klebsiella pneumoniae,* O-ag appears to stabilize capsule retention to increase serum resistance [67]. Future studies will be required to determine the role of L O-ag in capsule retention in *S.* Typhi and its contribution to serum resistance. Serum bactericidal assays are commonly performed with *Salmonella* cultured in LB medium. Here, we show clear differences in the role of O-ag length in conferring serum resistance to nontyphoidal and enteric fever *S. enterica* serovars cultured in LB. Future studies should also analyze serum resistance of different serovars under conditions that mimic the *Salmonella*-containing vacuole, which has been shown to affect O-ag length and resistance to complement killing by increasing the expression of complement-cleaving proteases [68,69].

In summary, we have systematically compared the roles of L and VL O-ag and Vi capsule in the interaction of serum complement and antibody with nontyphoidal and enteric fever *Salmonella* serovars. This first contact between pathogen and host immune effectors has critically important implications for subsequent pathogenic events. All three *Salmonella* serovars examined in this study resist serum killing but employ different mechanisms to do so (Fig 8). Nontyphoidal *S.* Typhimurium avidly binds complement and allows complement activation, leading to an inflammatory response that it withstands and exploits. In contrast, the enteric fever serovars *S.* Typhi and *S.* Paratyphi employ distinct but functionally analogous mechanisms to resist complement and antibody binding and prevent complement activation, which allows them to evade innate immunity and disseminate via trafficking phagocytes. It has been suggested that *S.* Typhi lost VL O-ag production and acquired the Vi capsule in response to the selective pressure imposed by serum components including complement and immunoglobulins [18,32,70]. Similarly, selection and convergent evolution have driven *S.* Paratyphi A to overexpress VL O2-antigen as an analogous barrier to complement, immunoglobulins and phagocytes [20]. We posit that the same selective pressures favored the R98C WzzB mutation in *S.* Paratyphi A to reduce L O-ag production, which reduces IgM binding and increases VL O-ag production. *S.* Paratyphi A VL O-ag production has been further enhanced

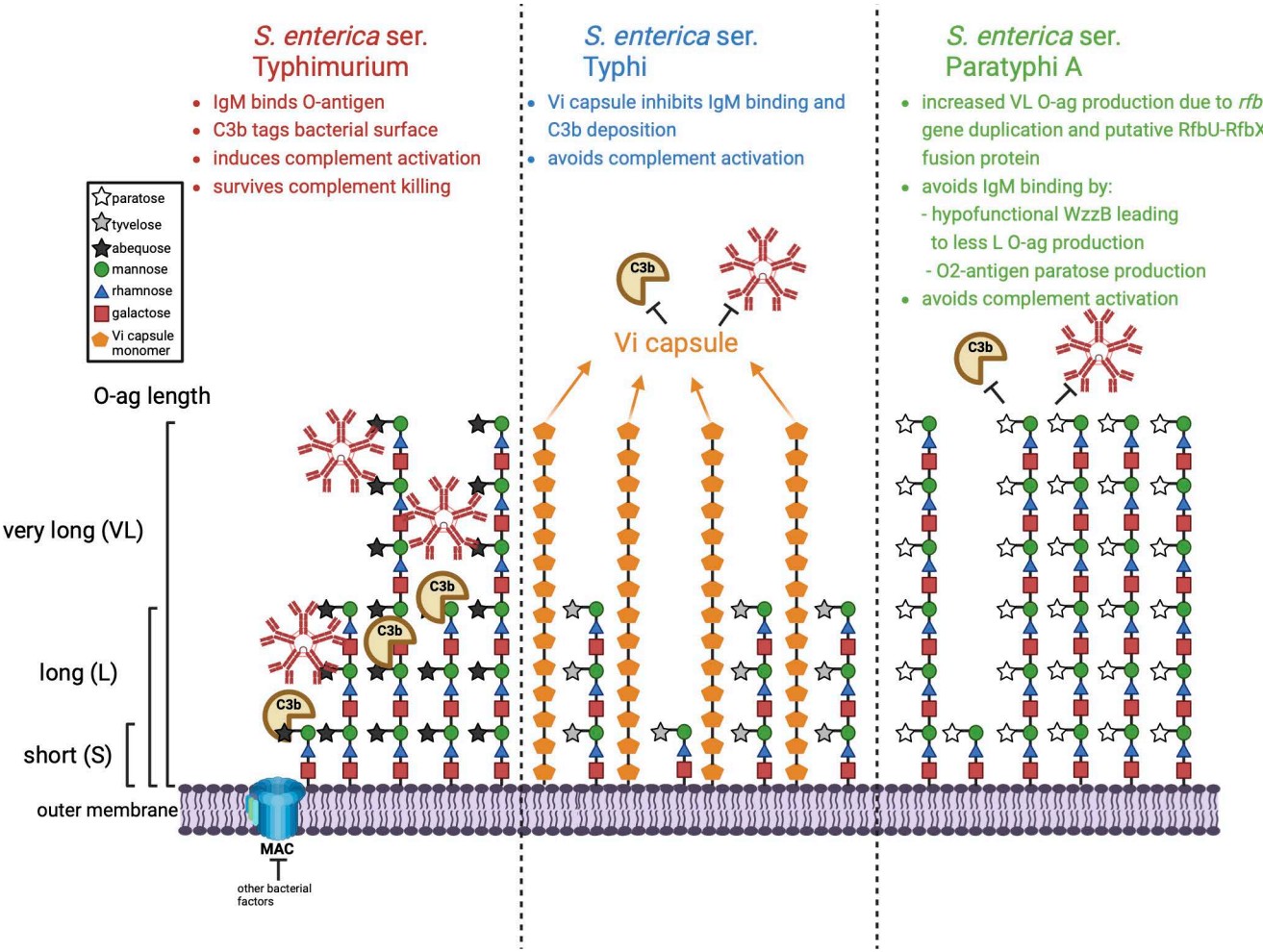

**Fig 8. Model of serum resistance in *Salmonella enterica* nontyphoidal serovar Typhimurium and the enteric fever serovars Typhi and Para-typhi A.** *S.* Typhimurium binds serum antibodies and complement inducing a pro-inflammatory response but survives serum bactericidal mechanisms. The enteric fever *Salmonella* serovars have evolved to evade host recognition by convergent pathways. The *S.* Typhi Vi capsule shields against complement deposition and antibody binding. *S.* Paratyphi A, which lacks the Vi capsule, has acquired several mutations that increase the amount of VL O-ag to protect against complement deposition and antibody binding. These include duplication of the *rfbV* glycosyltransferase, a point mutation in WzzB that inhibits antibody binding and L O-ag production and further increases VL O-ag production, and O2-antigen paratose production to decrease IgM binding. Created in BioRender. Guerra, F. (2025) https://BioRender.com/b51q154.

by *rfbV* gene duplication and creation of the partial RfbU-RfbX fusion protein, which reduce complement and IgM binding. Thus, the enteric fever *Salmonella* serovars have evolved to evade host recognition by convergent pathways, in contrast to the pro-inflammatory nontyphoidal serovar *S.* Typhimurium. These differences play an important role in the distinctive clinical manifestations of enteric fever and *Salmonella* enteritis.

## Materials and methods

### Bacterial strains and growth conditions

Bacterial strains used in this study are listed in S1 Table. Mutant alleles were constructed by λ-Red recombination, as previously described [71–73]. Allelic exchanges were constructed by λ-Red recombination with positive

selection for loss of tetracycline resistance, as previously described, with adjusted concentration of fusaric acid to 9 µg mL$^{-1}$ to select for *S.* Paratyphi A [71]. Three non-synonymous single nucleotide polymorphisms (SNPs) in *S.* Paratyphi A *wzzB* were separately introduced into *S.* Typhi *wzzB* cloned into pBlueScript II SK (+) (pBSIISK, Stratagene), using the QuikChange Lightning kit (Agilent, catalog #210518). The plasmid pBSIISK_*wzz*B$_{Typhi}$ with the introduced SNPs was used as template for PCR amplification and allelic exchange of *wzzB* in *S.* Paratyphi A. Plasmids and oligonucleotides used in this study to construct bacterial strains are listed in S2 Table. Mutant alleles were confirmed by PCR and allelic exchanges confirmed by sequencing. *S. enterica* cultures were grown in Luria-Bertani Broth (LB; Fisher Scientific) at 37 °C with shaking at 250 rpm. Length of incubation is detailed in subsequent sections.

### Construction of the *S.* Paratyphi A strain with a single gene copy of *rfbX, rfbV, and rfbU*

Primers FEGP107 and FEGP108 were used to amplify *tetRA* from genomic DNA of *S.* Typhimurium containing a *Tn10dTc*[del-25] insertion (strain JK18). The amplicon was electroporated into *S.* Paratyphi A strain ATCC 9150 to replace the *rfbX-rfbU* genomic region with *tetRA* by λ-Red homologous recombination. The *rfbX-rfbU* region from *S.* Typhi strain Ty2 was amplified using primers FEGP131 and FEGP132 and ligated to 900 bp upstream (amplified using primers FEGP129 and FEGP130) and downstream (amplified using primers FEGP131 and FEGP132) amplicons containing homology arms to the corresponding genomic region in *S.* Paratyphi A. The amplicons were ligated into BamHI-HF and EcoRI-HF restriction enzyme-digested pFOK using NEBuilder HiFi DNA Assembly (New England Biolabs) and electroporated into donor *E. coli* strain JKe201, as described [74]. The donor *E. coli* strain containing pFOK with *rfbX-rfbU* from *S.* Typhi was mated with *S.* Paratyphi A Δ*rfbX-U*::*tetRA* to promote allelic exchange of the *S.* Typhi *rfbX-rfbU* genes into *S.* Paratyphi A, as described [74]. A similar strategy was then used to sequentially replace the inserted *S.* Typhi *rfbX, rfbV,* and *rfbU* genes with the corresponding *S.* Paratyphi A genes. Each inserted *S.* Typhi gene on the *S.* Paratyphi A chromosome was replaced by *tetRA* using λ-Red recombination and then replaced with the *S.* Paratyphi A homologous gene by conjugation with *E. coli* JKe201 containing the pFOK plasmid with the replacement gene flanked by homology recombination regions. After a *S.* Paratyphi A strain with single copies of *rfbX, rfbV,* and *rfbU* was constructed, primers FEGP129 and FEGP134 were used to amplify the *rfbX-rfbU* region containing flanking homology regions, ligated into pFOK, and transformed into *E. coli* JKe201, before finally conjugating into *S.* Paratyphi A Δ*rfbX-U*::*tetRA* for allelic exchange of the *rfbX-rfbU* region in one step. Gene constructs were confirmed by sequencing. To construct constitutively expressed *rfbV* for complementation in *S.* Paratyphi A, primers FEGP254 and FEGP255 were used to amplify *rfbV* from *S.* Paratyphi A gDNA, and the amplicon was ligated into pJK770 that was digested with NcoI-HF using NEBuilder HiFi DNA Assembly. Similarly, *fepE* and *rfbV* from *S.* Typhimurium, and *rfb*U*XV* and *rfb*U*X* from *S.* Paratyphi A were amplified using primers in S2 Table and ligated into pJK770.

### Serum bactericidal activity

Bacteria were grown in 5 mL LB at 37 °C with shaking at 250 rpm for 18 hrs to reach stationary phase. Bacterial optical density at 600 nm (OD$_{600nm}$) was adjusted to 1.0, bacteria were pelleted by centrifugation, supernatant was discarded, and pellets were resuspended in PBS to ~1 x 10$^9$ CFUs mL$^{-1}$. Two-hundred µL of washed bacteria were mixed with 200 µL of human pooled serum (MP Biomedicals, catalog #2930149) in a 1.5 mL Eppendorf tube to attain a 50% serum concentration. For bactericidal experiments in 25% human pooled serum, 200 µL of washed bacteria were mixed with 200 µL of 50% human pooled serum diluted in PBS. Bacteria and serum mixtures were incubated at 37 °C for 1 hr. Following incubation, 20 µL of bacteria-serum mixture were added to 180 µL of ice-cold PBS and serial dilutions plated on LB agar. CFU were enumerated following overnight incubation at 37 °C. Serum killing was calculated relative to bacteria incubated in parallel with PBS instead of human pooled serum.

### C3b deposition and immunoglobulin binding

Bacteria were incubated in 50% human pooled serum, as described above. After 20 min incubation, 100 µL of bacteria-serum mixture were added to 1 mL ice-cold PBS, pelleted by centrifugation, and supernatant was discarded. Bacteria were incubated with mouse phycoerythrin anti-human C3b/iC3b (BioLegend, catalog #846104), mouse Alexa Fluor 488 anti-human IgG Fc (BioLegend, catalog #409322, discontinued; BioLegend, catalog #366921, can be used as an alternative clone), or mouse Alexa Fluor 488 anti-human IgM (BioLegend, catalog #314533) for 30 min on ice. Unbound antibody was washed twice with 1 mL PBS by pelleting bacteria before fixing with 100 µL IC Fixation Buffer (Invitrogen, catalog #00-8222-49) for 20 min at room temperature. Fixed bacteria were washed with 1 mL PBS and resuspended in PBS. Analysis was performed on 50,000 events collected in a BD LSR II Flow Cytometer or FACSymphony A3 Cell Analyzer with low sample flow rate.

### ELISA (Enzyme-linked immunosorbent assay)

Bacteria were incubated in 50% human pooled serum as described above. After 60 min incubation at 37 °C, bacteria were pelleted by centrifugation at 20,000x$g$ for 3 min at 4 °C. The top 200 ul of supernatant were removed and stored at -80 °C until analysis. Human C3a and C5a production were measured using the Human Complement C3a ELISA Kit (Invitrogen, catalog #BMS2089) and the Human C5a ELISA Kit (Invitrogen, catalog #BMS2088), respectively, following the manufacturer's instructions.

### LPS extraction, staining, and densitometry analysis

LPS extraction was performed as previously described [75]. LPS was separated in a Novex WedgeWell 14% Tris-Glycine Gel (Invitrogen, catalog #XP00145BOX). Silver staining of extracted LPS was performed as previously described [76]. Imaged gels were analyzed with Fiji [77] to obtain area under the curve of the lipid A+core and VL O-ag bands.

### Statistical analyses

Statistical comparisons were performed using Prism version 9.2.0 (GraphPad). Statistical method and sample size for experiments are detailed in the figure legends. Flow cytometry data were analyzed using FlowJo version 10.7.1 (Beckton Dickinson & Company).

## Supporting information

**S1 Fig. Constitutive *rfbV* expression in *S.* Typhimurium does not increase VL O-ag production.** (A) LPS extraction followed by gel electrophoresis and silver staining shows that constitutive *rfbV* expression in *S.* Typhimurium does not increase VL O-ag production nor (B) decrease C3b deposition. STm, *S.* Typhimurium.
(TIF)

**S1 Table. Bacterial strains and plasmids used in the study.**
(PDF)

**S2 Table. Primers used in the study.**
(PDF)

**S1 Data. Data values for figures.**
(XLSX)

## Acknowledgments

Research reported in this publication was generated using the DLMP Flow Cytometry Core at the University of Washington. The authors are grateful to Dr. Mariya T. Sweetwyne for access to FluorChem Q imager.

## Author contributions

**Conceptualization:** Fermin E. Guerra, Joyce E. Karlinsey, Stephen J. Libby, Ferric C. Fang.

**Formal analysis:** Fermin E. Guerra, Ferric C. Fang.

**Funding acquisition:** Fermin E. Guerra, Ferric C. Fang.

**Investigation:** Fermin E. Guerra.

**Methodology:** Fermin E. Guerra, Joyce E. Karlinsey, Stephen J. Libby, Ferric C. Fang.

**Project administration:** Fermin E. Guerra, Ferric C. Fang.

**Supervision:** Fermin E. Guerra, Ferric C. Fang.

**Validation:** Fermin E. Guerra, Ferric C. Fang.

**Visualization:** Fermin E. Guerra, Joyce E. Karlinsey, Stephen J. Libby, Ferric C. Fang.

**Writing – original draft:** Fermin E. Guerra, Ferric C. Fang.

**Writing – review & editing:** Fermin E. Guerra, Joyce E. Karlinsey, Stephen J. Libby, Ferric C. Fang.

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
