## [Decision Letter · Decision Letter 0]

4 Mar 2025

PPATHOGENS-D-25-00152

Evasion of serum antibodies and complement by *Salmonella* Typhi and Paratyphi A

PLOS Pathogens

Dear Dr. Fang,

Thank you for submitting your manuscript to PLOS Pathogens. We feel that this is an interesting study that requires minor modification before publication. We invite you to submit a revised version of the manuscript that addresses the points raised during the review process. In particular, please make the text accessible to a more general audience and enhance readability by appropriate referencing of figures throughout. 

Please submit your revised manuscript within 30 days May 03 2025 11:59PM. If you will need more time than this to complete your revisions, please reply to this message or contact the journal office at plospathogens@plos.org. Please include the following items when submitting your revised manuscript:

We look forward to receiving your revised manuscript.

Kind regards,

Stephen J McSorley

Guest Editor

PLOS Pathogens

D. Scott Samuels

Section Editor

PLOS Pathogens

Sumita Bhaduri-McIntosh

Editor-in-Chief

PLOS Pathogens

orcid.org/0000-0003-2946-9497

Michael Malim

Editor-in-Chief

PLOS Pathogens

orcid.org/0000-0002-7699-2064

**Journal Requirements:**

- ® on pages: 27, and 28

- TM on page: 28.

3) Please amend your detailed Financial Disclosure statement. This is published with the article. It must therefore be completed in full sentences and contain the exact wording you wish to be published.

**Reviewers' Comments:**

Reviewer's Responses to Questions

**Part I - Summary**

Reviewer #1: The manuscript aims to elucidate the contribution of the O-antigen and the Vi capsule to serum resistance and complement deposition, by comparing three different Salmonella serovars: the non-typhoidal serovar Typhimurium, and the two enteric fever serovars Typhi and Paratyphi A. The study has the merit to meticulously dissect the similarities and differences in how three different Salmonella serovars resist or subvert immune complement and Ig binding. In particular, the data showing how multiple gene copies of rfbV in Paratyphi A alter LPS and serum resistance are important and convincing. The manuscript is overall clearly written, and the study is well conducted, but there are a few points that need to be addressed.

Reviewer #2: The interaction between the Salmonellae that cause systemic disease (and those that typically don’t) with the humoral immune system is something that warrants a much greater understanding. This is in part because of the plethora of new typhoid vaccines which are approved and WHO pre-qualified based on immunogenicity data, in the absence of a strict correlate of protection. Equally, a deeper understanding of how non-encapsulated S. Paratyphi appear to induce a very similar disease to that caused by S. Typhi but where the Paratyphi are without the major vaccine target and virulence determinant i.e. the Vi capsule, needs to be established and new paratyphoid vaccines move through development.

The paper addresses the role of the various forms of LPS in resistance to killing by complement. I was a little disappointed not to see what happens when a pooled immune serum is used and whether the responses observed carry onto circumstances where specific antibody is present.

These are key questions and while the paper could be made slightly more accessible e.g. bring Supp figure 2 cartoon’ into the manuscript, and some well targeted abbreviations used throughout, the manuscript addresses a question that I would see as suitable for PLoS Pathogens.

The quality of the data is generally very good, the interpretation is deep and, with some editing, the paper should make a significant contribution to our understanding of how Salmonellae engage with the naïve human humoral immune system.

Reviewer #3: In this manuscript (PPATHOGENS-D-25-00152), Guerra et al. examine the contributions of LPS O-antigen and the S. Typhi Vi capsule to serum resistance and complement activation in nontyphoidal and enteric fever serovars of Salmonella enterica. The study, which expands on earlier observations, systematically compares the contributions of long O-antigen and very long O-antigen to serum resistance and complement activation in S. Typhimurium, S. Typhi, and S. Paratyphi. The authors principally use wild-type and mutant strains lacking WzzB (long O-antigen chain length determinant), FepE (very long O-antigen chain length determinant), and/or the Vi capsule, and measure serum bactericidal activity, complement deposition, immunoglobulin binding, and anaphylatoxin release. They also use mutant strains of S. Typhimurium and S. Paratyphi lacking specific O-antigen-modifying glycosyltransferases and mutant strains of S. Paratyphi carrying different copies of rfbV, which encodes another type of O-antigen-modifying enzyme. The study provides compelling evidence indicating that S. Typhimurium, S. Typhi, and S. Paratyphi interactions with innate immune effectors fundamentally differ, yielding new insights into the mechanisms by which these serovars evade innate immune responses. Overall, the manuscript is well written and easy to read, the experiments are rigorously designed and well-controlled, and the results are clearly described and carefully interpreted.

**Part II – Major Issues: Key Experiments Required for Acceptance**

Reviewer #1: The points below do not necessarily require new experiments but nevertheless need to be considered based on the data and the controls provided.

1) Line 207-210: “Increased IgM binding to STy occurs in the absence of Vi capsule, but not in strains lacking both the Vi capsule and long O-antigen, suggesting that IgM binds to long O-antigen in the absence of Vi capsule.” The conclusion does not appear to be fully supported by the data presented. IgM binds STy, STy wzzB, and STy wzzBvexA to similar degree and they are all higher than the unstained control (Fig 1E). This would suggest that IgM is not exclusively binding long O-antigen unless the ~250 MdFI is non-specific. To address the possibility of non-specific binding, flow cytometry needs a proper isotype control. For the PE anti-C3b/iC3b antibody, BioLegend has an isotype control (Cat # 400113). For AF488 anti-human IgM (Cat#400132). The catalog # listed for the anti-human IgG may be a typo.

2) Line 289-290: “Previous studies have shown that the composition of O-antigen can impact C3b deposition [46-48].” Ref 46 and 48 are about the alternative pathway of complement activation. For this manuscript, mechanistically, is this classical, alternate, or lectin pathway activation of complement? If classical, is this IgM- or IgG-dependent? In Figure 4A-D, IgM binding does not strongly correlate with C3b deposition. IgG binding MdFI seems to match %C3b+ but not C3b deposition. Related, deletion of OafA in STm didn’t impact C5a release but increased IgM binding (Fig 6F, G). This suggests that Ig binding and complement binding are not completely in sync and therefore not entirely the classical pathway. Complement technology has various complement depleted-human sera such as C1q-depleted serum (Cat#A300) that could help address this. It is likely a mix of pathways for all serovars but it’s possible that one serovar may lean more towards the alternative pathway. Even if complement deposition is primarily due to alternative pathway, IgG and IgM binding still opsonize the bacteria and could be discussed.

Reviewer #2: There are no major issues - in Minor Issues I have raised some methodological issues. I the key Figure 2A, it is difficult for me to see the detail.

Reviewer #3: None.

**Part III – Minor Issues: Editorial and Data Presentation Modifications**

Reviewer #1: 3) Figure 1A: If 10^-8 is the limit of detection, please mark it as such.

4) Fig 1A, B and Fig 7E are the only figures referenced in the discussion although data of the entire manuscript is discussed. Please update referenced figures for consistency in the text.

5) It is not clear in the figure legends if “independent” is referring to independent experiments or biological replicates.

6) While serum bactericidal assays are commonly done with Salmonella cultured in LB, media mimicking the SCV has been shown to alter O-antigen of Salmonella Typhimurium (PMID 29866904). Could the authors add some discussion about this topic.

Reviewer #2: Comments and Questions

1. Abstract – long and very long O antigen polymers need to be more accurately described and a suitable abbreviation found. The number of repetitive sugars, or apparent MW in SDS could be used to define ranges for each of the pathogens. Given the plasticity in chain length and loose control over polymerisation rate (which yields something of a normal distribution) getting the chemical differences between long and very long out of the way early, would make the paper more readable. Importantly, the research question(s) is clear.

2. The introduction is overly detailed. Some of this detail can be referenced or, where relevant, placed in the Discussion. While the mechanism through which complement works is important, ultimately the study focusses on C3b deposition. A cartoon of the different O antigens for the 3 serovars under examination would be helpful, i.e. where the differentiating sugars like tyvelose sit in the LPS.

3. Results – In Figure 1, it would have been good to see some more conclusive link between the killing observed and complement or natural antibody or both. This could be done with low technology (heating serum) or specific complement removal and/or addition to say heated serum. This is an important Figure insofar as the rest of the paper builds from there. It would have been interesting to see how pooled immune sera reacted with the various forms of LPS and whether there was equivalent killing of the mutants. Granted, adjusting for the isotypes and specificities of the antibodies raised by paratyphoid, Typhimurium or typhoid infection would be difficult to QC, but this data may have provided some insights into how the vaccines work.

4. In S. Typhi, it is possible that there is a link between O side chain length and Vi encapsulation, as has been reported for pathogenic Klebsiellae (10.1128/spectrum.01517-21). This means that a mutation in O length in S. Typhi could also result in de-encapsulation. Some comment on this or measurement of the capsule in the WzzB might be important.

5. In Figure 2, my copy of the figure made it very hard to see the VL Oag. I don’t know if this can be enhanced. It is hard to tell whether the levels are going up or down or staying the same. The abbreviations of the LPS forms used Figure 2 are suitable for use in the whole paper.

6. The relative binding of the different antibody types is a very fraught experiment when unfractionated serum is used because there will be competition between high and low affinity antibodies of the different types. This means that a reading where IgG>IgM can be explained in many different ways – it can be because the IgG has greater affinity displacing IgM, or it is present in higher levels enabling competitive binding. The reagents used to detect Ig binding by antibody type may also have different affinities and hence sensitivities. The only fundamental way to determine binding is to use a monoclonal antibody that is directly conjugated and standardised. This is not a suggestion, but possibly is a limitation.

7. While the binding of C3b is done well, the question that might be asked is whether the C3b ‘on rate’ is equal where it binds to VL and L LPS, and/or whether the ‘off rate’ is equal for the different LPS types, i.e. whether the apparent differences in association of C3b represents altered kinetics. Is the stability of VL LPS the same as long LPS – i.e. would C3b bound to VL LPS be more readily shed during the analysis? There is equipment to measure binding parameters (e.g. BIACore) but it adds a complexity that is probably unnecessary. Could the binding kinetics explain differences between Paratyphi and Typhi (line 444) which may also have Vi capsule diffusion/penetration to contend with. Is there a model to determine whether the VL LPS extends beyond Vi capsule in S. Typhi, or is it present only below the capsule surface? It is possible to measure capsule thickness using AFM (10.1128/spectrum.01517-21). If C3b binds to VL LPS below the capsule surface then its ability to act as an opsonin will will be greatly diminished because the capsule could provide stearic hindrance.

8. Another comment that is not easily answerable. Do we know whether the VL and L LPS is equally distributed over the surface of the bacterium and does this matter? Are there likely to be ‘rafts’ of increased expression of the VL form (e.g. poles) or is the surface expression equally distributed – this is a difficult question to answer because there are ? no reagents that can stain VL and L forms of LPS.

9. Discussion – this section is done well. The ‘even bigger’ perspective might be important here. Three evolutionarily linked serovars having evolved different mechanisms for dealing with the binding of non-immune antibody and complement to the bacterial surface. It might be reflected somewhere that S. Typhimurium is really only an accidental pathogen of humans – the transmission chains that are well mapped for S. Typhi and less clear for human infections of iNTS. If S. Typhimurium spends the majority of its existence outside of the human host (cf. S. Typhi), should we expect the relationships with human antibody to be the same or different. As importantly, how does S. Paratyphi that lack a key virulence determinant (Vi) produce a syndrome that looks like typhoid fever – from this perspective, using a Vi-encapsulated S. Paratyphi may have been instructive, and to test whether there WzzB-driven approaches to serum resistance are still observed in S. Paratyphi that carry Vi.

Reviewer #3: 1. The finding that rfbV gene duplication contributes to serum resistance in S. Paratyphi is interesting, yet the mechanism by which it enhances the quantity of very long O-antigen production remains incompletely understood. Is it simply a gene dosage effect? Are the three copies of rfbV identical? What is role of the hypothetical rfbU-rfbX fusion ORF in O-antigen production and how does it affect serum resistance? The authors present an intriguing hypothesis that could easily be tested, or so it seems.

2. More careful consideration should be given to the use of genetic complementation of select mutations or gene deletions.

PLOS authors have the option to publish the peer review history of their article (what does this mean? ). If published, this will include your full peer review and any attached files.

**Do you want your identity to be public for this peer review?** For information about this choice, including consent withdrawal, please see our Privacy Policy .

Reviewer #1: No

Reviewer #2: **Yes: ** Richard Strugnell

Reviewer #3: No

**Figure resubmission:**
---

## [Decision Letter · Decision Letter 1]

18 Apr 2025

Dear Dr. Fang,

We are pleased to inform you that your manuscript 'Evasion of serum antibodies and complement by *Salmonella* Typhi and Paratyphi A' has been provisionally accepted for publication in PLOS Pathogens.

Best regards,

Stephen J McSorley

Guest Editor

PLOS Pathogens

D. Scott Samuels

Section Editor

PLOS Pathogens

Sumita Bhaduri-McIntosh

Editor-in-Chief

PLOS Pathogens

orcid.org/0000-0003-2946-9497

Michael Malim

Editor-in-Chief

PLOS Pathogens

orcid.org/0000-0002-7699-2064

Reviewer Comments (if any, and for reference):

Reviewer's Responses to Questions

**Part I - Summary**

Reviewer #1: The authors have fully addressed my prior comments and I support the publication of this important work.

Reviewer #2: This is a re-review. The paper addresses an important and interesting question about antibody and complement binding in different Salmonella enterica, and the roles of capsules and LPS in these phenomena. It is well executed and thoroughly interpreted.

Reviewer #3: The authors adequately addressed the reviewers' comments.

**Part II – Major Issues: Key Experiments Required for Acceptance**

Reviewer #1: (No Response)

Reviewer #2: No major issues - these have been addressed in the re-write.

Reviewer #3: (No Response)

**Part III – Minor Issues: Editorial and Data Presentation Modifications**

Reviewer #1: (No Response)

Reviewer #2: There are no remaining minor issues - these have been adequately rebutted.

Reviewer #3: (No Response)

PLOS authors have the option to publish the peer review history of their article (what does this mean? ). If published, this will include your full peer review and any attached files.

**Do you want your identity to be public for this peer review?** For information about this choice, including consent withdrawal, please see our Privacy Policy .

Reviewer #1: No

Reviewer #2: **Yes: ** Dick Strugnell

Reviewer #3: No

---

## [Editor Report · Acceptance letter]

Dear Dr. Fang,

We are delighted to inform you that your manuscript, "Evasion of serum antibodies and complement by *Salmonella* Typhi and Paratyphi A," has been formally accepted for publication in PLOS Pathogens.

Best regards,

Sumita Bhaduri-McIntosh

Editor-in-Chief

PLOS Pathogens

orcid.org/0000-0003-2946-9497

Michael Malim

Editor-in-Chief

PLOS Pathogens

orcid.org/0000-0002-7699-2064